# Trajectories of self-rated health in an older general population and their determinants: the Lifelines Cohort Study

Marlies Feenstra [ID],[1] Barbara C van Munster,[1,2] Janet L MacNeil Vroomen,[3,4] Sophia E de Rooij,[1] Nynke Smidt[1,5]

For numbered affiliations see end of article.

**Correspondence to**
Marlies Feenstra;
m.feenstra01@umcg.nl

## ABSTRACT

**Objectives** Poor self-rated health (SRH) is a strong predictor of premature mortality in older adults. Trajectories of poor SRH are associated with multimorbidity and unhealthy behaviours. Whether trajectories of SRH are associated with deviating physiological markers is unclear. This study identified trajectories of SRH and investigated the associations of trajectory membership with chronic diseases, health risk behaviours and physiological markers in community-dwelling older adults.

**Study design and setting** Prospective general population cohort.

**Participants** Trajectories of SRH over 5 years were identified using data of 11 600 participants aged 65 years and older of the Lifelines Cohort Study.

**Outcome measures** Trajectories of SRH were the main outcome. Covariates included demographics (age, gender, education), chronic diseases, health-risk behaviour (physical activity, smoking, drinking) and physiological markers (body mass index, cardiovascular function, lung function, glucose metabolism, haematological condition, endocrine function, renal function, liver function and cognitive function).

**Results** Four stable trajectories were identified, including excellent (n=607, 6%), good (n=2111, 19%), moderate (n=7677, 65%) and poor SRH (n=1205, 10%). Being women (OR: 1.4; 95% CI: 1.0 to 1.9), low education (OR: 2.1; 95% CI: 1.5 to 3.0), one (OR: 10.4; 95% CI: 7.4 to 14.7) or multiple chronic diseases (OR: 37.8; 95% CI: 22.4 to 71.8), smoking (OR: 1.8; 95% CI: 1.0 to 3.2), physical inactivity (OR: 3.1; 95% CI: 1.8 to 5.2), alcohol abstinence (OR: 2.2; 95% CI: 1.4 to 3.2) and deviating physiological markers (OR: 1.5; 95% CI: 1.1 to 2.0) increase the odds for a higher probability of poor SRH trajectory membership compared with excellent SRH trajectory membership.

**Conclusion** SRH of community-dwelling older adults is stable over time with the majority (65%) having moderate SRH. Older adults with higher probabilities of poor SRH often have unfavourable health status.

## BACKGROUND

Self-rated health (SRH) is known as an inclusive measure of global health and is often used as a supplement to objective clinical measures of physical health such as presence

### Strengths and limitations of this study

► This study concerns the evaluation of physiological markers as a determinant of self-rated health trajectories.
► The study results are representative for Dutch community-dwelling adults aged 65 years and older.
► Reverse causation could not be eliminated.
► The number of chronic conditions were based on self-report; this could have caused non-differential misclassification bias.

of disease and disability.[1 2] In older adults, poor SRH is an independent and strong predictor of premature mortality.[3 4] However, evidence for factors associated with poor SRH is predominantly cross-sectional and longitudinal evidence is required. Analysis of latent clusters of individuals who follow a similar pattern of SRH over time, so-called trajectory analysis, can be used to explore the course of SRH in time within a certain population.[5] Few studies have studied SRH in community-dwelling older adults by trajectory analysis revealing various numbers of identified trajectories.[6–8] Distinct trajectories of SRH varied from persistently good,[6 7] persistently moderate,[7 8] persistently poor,[6 7] declining[6–8] to improving trajectories of SRH.[6] People in declining SRH trajectories were differentiated at baseline by older age, lower education level and an increased number of chronic conditions compared with people in consistently good SRH trajectories.[6–8] However, in these studies, other measures of determinants of health status such as abnormalities in physiological markers, like blood pressure, thyroid hormone levels and glycated haemoglobin (HbA1c) were not evaluated. Such markers reflect cross-sectional clinical parameters of physiological processes.[9] Abnormal physiological processes may indicate pre-clinical

BMJ

prodromal phases of underlying diseases which are suggested to play a role in burden of disease expressed by poor SRH evaluations in older adults.[1 4 10 11] We hypothesise that multimorbidity, health risk behaviours, and deviations in physiological markers are associated with trajectories that lead to poor SRH.

The aim of this study is to identify classes of SRH over 5 years in community-dwelling older adults and to investigate whether group membership of SRH trajectories is associated with self-reported chronic diseases, health risk behaviours and physiological markers.

## METHODS

### Study population
A subsample of the adult Lifelines Cohort Study was used, including participants aged 65 years or older at baseline (n=12 685). A detailed description of the complete Lifelines cohort profile is described elsewhere.[12]

### Measurements
#### Primary outcome measure
Repeated measures of SRH were assessed at baseline, 1.5 years, 3 years and 5 years after baseline measurement by means of a self-reported question 'how would you rate your health in general? (excellent, very good, good, fair, poor)'.[13 14] The single item SRH question with five response options is a valid and reliable measure of general health status in older adults.[15–17]

#### Covariates
Demographics included *age, sex* and *education level* (low, less than primary through lower secondary; intermediate, upper secondary through post-secondary, non-tertiary; high, short cycle tertiary and higher[18 19]).

*Chronic diseases* were categorised (none, one, two or more) based on a participant's baseline report on presence of the most burdensome chronic diseases as forecasted for the next decades by RIVM,[20] including dementia, myocardial infarction, osteoarthritis, cerebrovascular accident, diabetes, chronic obstructive pulmonary disease (COPD), cancer, anxiety and mood disorders.

Health risk behaviours included *physical activity* (≥5, 2–4, ≤2 days/week physically active for at least 30 min[21]), *smoking* (never, former, current smoker), *alcohol consumption* (abstainer, low risk, at risk[22]). Low risk drinking is defined as no more than three drinks per day and no more than seven drinks per week for both women and men.[23]

Physiological markers included body mass index (BMI) as a marker of body composition[24 25]; systolic and diastolic blood pressure were interpreted with total cholesterol (CHOL) and high-density lipoprotein (HDL) ratio as a marker of cardiovascular function[24]; forced expired volume in one second and the forced vital capacity ratio were used as a marker of lung function[26 27]; HbA1c as a marker of glucose metabolism[24 28]; total haemoglobin (Hb) as a marker of haematological condition[28]; thyroid

stimulating hormone and free thyroxine were used as markers of endocrine function[29–31]; estimated glomerular filtration rate using the Cockcroft Gault formula was used as a marker of renal function[32–34]; hepatic steatosis index (HSI) was used as a marker of liver function[35 36]; and the mini-mental state examination score was used as a marker of cognitive function.[24 37] A detailed description of physiological markers used and clinical cut-offs are presented in online supplementary appendix A table A1. Based on clinical cut-offs, both *individual physiological markers* (normal, abnormal values) and a *sum score of abnormal physiological markers* were used in the analyses (<3 vs. ≥3 abnormal physiological markers).

### Statistical analyses
Baseline characteristics of all participants and classified by SRH trajectory groups were expressed in median and IQR for continuous variables and proportions and percentages for categorical variables. To identify distinct trajectories of SRH over 5 years, latent class analyses were performed using group-based trajectory modelling.[38] The trajectory model was built by a stepwise approach:

Step 1: The basic model was build by including the four repeated measures of SRH using a censored normal model. Two to six trajectories were considered after which the optimal number of trajectories was selected using highest Bayesian information criterion (BIC),[39] and Bayes factor.[40] After the optimal number of trajectories was determined, optimal trajectory shape was determined by varying the growth terms. Optimal trajectory shape was evaluated based on (1) the probability of a person belonging to the selected trajectory (>0.7), (2) the odds of correct classification (>5.0), (3) close correspondence between the estimate of group membership probability and the proportion of individuals classified to the group and (4) reasonable narrow CIs for the estimates of group membership probability.[41] For the latter two no formal criteria for maximum deviation were available.

Step 2: Multivariable multinomial logistic regression analyses were performed to estimate associations between the probability of SRH trajectory group assignment (result of step 1) and covariates. Three theoretical models were investigated. Model 1: chronic diseases and health behaviours; Model 2: model 1 plus physiological markers; Model 3: model 1 plus the sum score of abnormal physiological markers. For all determinants, multicollinearity was checked using Pearson's correlations. Baseline age, sex and level of education were included in all models. Model selection was based on lowest BIC, and Akaike's information criterion (AIC).[42]

Step 3: Trajectories of SRH were re-estimated by including the covariates of the selected model out of step 2. This last step allows to evaluate the influence of one covariate on the probability of belonging to each trajectory taking into account the uncertainty of posterior group membership probability that is introduced by trajectory analysis. Wald statistics were applied for

testing the differences between covariates across trajectory groups.

Data of participants with missing data of SRH at all time points were excluded from all analyses (n=1085 (9%)). Participants with missing SRH data at three or less time points were handled using maximum likelihood estimation. Maximum likelihood estimation uses all available information from observed data for constructing the likely values for missing data.[41] From step 2 onwards, participants who had missing data for baseline covariates were excluded from further analyses (n=3010 (26%)). The flow of participants from the initial to the analytic sample is presented in online supplementary appendix B and figure B1.

Sensitivity analyses were performed by: (1) rerunning basic trajectory analysis accounting for non-random attrition (dual trajectory modelling), and (2) using a composite score for chronic diseases without anxiety and mood disorders. For all analyses Stata Statistical Software release 14 was used (StataCorp. 2015) with the traj plug-in.[43 44]

## RESULTS
### Study population characteristics
Of all 11 600 participants, median age at baseline was 69 years (range 65–93), and 47% were men. Of this sample, 34% reported one chronic disease at baseline, 13% reported multimorbidity (≥2 chronic diseases), 57% had one or two abnormal physiological markers, and 38% had three or more abnormal physiological markers (table 1). Over 5 years of follow-up, 497 people died (4%), and 3721 (32%) were lost to follow-up. The 3010 (26%) participants who were excluded from the analysis in steps 2 and 3 due to missing covariates measured at baseline were older, more often women, lower educated and had relatively less self-reported chronic diseases, but more abnormal values of physiological markers compared with the participants retained in the analysis (completers) (table 2). One of the reasons for these missing data was that participant with low cognitive abilities (mini-mental state examination <26) had a shorter proxy interview, which was the case in 1261 (42%) of the excluded participants.

### Trajectories of SRH over 5 years
Of all evaluated models, four trajectories of SRH with different intercepts, and all slopes close to zero showed the best fit (fit statistics are presented in online supplementary appendix table C1 and C2). The four trajectories were identified as excellent, good, moderate and poor SRH including 607 (5.6%), 2111 (18.8%), 7677 (65.3%) and 1205 (9.6%) participants, respectively (figure 1; Appendix C Figure C1).

Table 1 presents baseline characteristics of participants in all trajectory groups. People having the highest probability of poor SRH trajectory membership were on average older, more often women, lower educated, more often physically inactive, more often alcohol abstainer

and they had more self-reported chronic diseases compared with people who have highest probabilities of assignment to the excellent, good and moderate SRH trajectories. Concerning objectively measured physiological markers, people having the highest probability of poor SRH trajectory membership had higher BMI, less often high blood pressure, but more often high CHOL/ HDL ratio, higher Hb levels, higher HSI index, and they scored lower on cognitive function compared with people with highest probability of assignment to moderate, good and excellent SRH trajectories. In addition, people with the highest probability for poor SRH trajectory membership had more abnormal values of physiological markers compared with people with highest probability of assignment to moderate, good and excellent SRH trajectories.

### Identification of covariates of trajectory membership probability
Table 3 presents the results from multivariate logistic regression analyses on probability of group membership of SRH. Model 2 performed worse compared with model 1 (BIC: −61 942; AIC:1.811). The simplest model with only self-reported covariates (model 1) had lowest BIC (−62 488), but higher AIC (1.807) compared with model 3 that included a sum score of physiological markers as well (BIC:−61 718; AIC: 1.804).

However, both models had different sample sizes due to missing values for physiological markers in model 3. Taking into account the exploratory nature of this step in the analysis, type II error (an underfit model) would be more undesirable than type I error (an overfit model). Therefore the covariates included in model 3 were used for the final model (see table 3, model 3).

### Final model adjusted for associated covariates
The final trajectory model was modelled by jointly estimating the basic model and the covariates age, sex, educational level, self-reported chronic diseases, physical activity behaviour, smoking behaviour, alcohol consumption and the sum score of affected physiological markers as risk factors. The final model assigned 471 (5.5%), 1716 (20.0%), 5637 (65.6%) and 766 (8.9%) people to the excellent, good, moderate and poor SRH trajectories. The final model including covariates showed best fit statistics of posterior probability of group assignment (Appendix D, Table D1). The basic model over-represented the proportion of participants with highest probability of poor and moderate SRH trajectory membership, and under-represented the proportion of people with highest probability of excellent and good trajectory membership, compared with the final model that took into account the effect of covariates (online supplementary appendix D and figure D1).

Table 4 presents the ORs of each of the evaluated covariates of people with highest probability of poor, moderate and good SRH trajectory membership using the excellent SRH trajectory as reference category. Increasing number of chronic diseases increased the odds of higher

**Table 1** Baseline characteristics of all participants aged 65 years and older and categorised by SRH trajectory group

| Characteristic | All | 1. Excellent | 2. Good | 3. Moderate | 4. Poor |
|---|---|---|---|---|---|
| n | 11 600 | 607 | 2111 | 7677 | 1205 |
| Demographics | | | | | |
| Age, median (IQR 25;75) | 69 (66; 73) | 68 (66; 72) | 69 (66; 72) | 69 (66; 72) | 70 (67; 74) |
| Range (years) | 65–95 | 65–90 | 65–92 | 65–93 | 65–90 |
| Missing | — | — | — | — | — |
| Sex, n (%) male | 5484 (47) | 344 (57) | 1161 (55) | 3523 (46) | 456 (38) |
| Missing | — | — | — | — | — |
| Highest level of education, n (%) | | | | | |
| Low | 6563 (57) | 301 (50) | 1006 (48) | 4482 (58) | 774 (64) |
| Intermediate | 2037 (18) | 107 (18) | 407 (19) | 1345 (18) | 178 (15) |
| High | 2239 (19) | 168 (28) | 592 (28) | 1319 (17) | 160 (13) |
| Missing | 761 (7) | 31 (5) | 106 (5) | 531 (7) | 93 (8) |
| Health status, n (%) | | | | | |
| SRH | | | | | |
| Excellent | 645 (6) | 373 (62) | 246 (12) | 26 (<1) | — |
| Very good | 2290 (20) | 155 (26) | 1326 (63) | 804 (10) | 5 (<1) |
| Good | 6358 (55) | 4 (<1) | 344 (16) | 5805 (76) | 205 (17) |
| Fair | 979 (8) | — | — | 275 (4) | 704 (58) |
| Poor | 20 (<1) | — | — | — | 20 (2) |
| Missing | 1308 (11) | 75 (12) | 195 (9) | 767 (10) | 271 (22) |
| Chronic diseases (self-reported) | | | | | |
| None | 6076 (52) | 467 (77) | 1386 (66) | 3871 (50) | 351 (29) |
| 1 | 3979 (34) | 116 (19) | 604 (29) | 2793 (36) | 466 (39) |
| ≥2 | 1545 (13) | 24 (4) | 121 (6) | 1013 (13) | 388 (32) |
| Missing | — | — | — | — | — |
| Health behaviours, n (%) | | | | | |
| Physical activity for at least 30 min | | | | | |
| ≥5 days/week | 6395 (55) | 368 (61) | 1330 (63) | 4226 (55) | 471 (39) |
| 2–4 days/week | 2481 (21) | 109 (18) | 396 (19) | 1743 (23) | 233 (19) |
| ≤1 day/week | 761 (7) | 27 (5) | 93 (4) | 512 (7) | 129 (11) |
| Missing | 1963 (17) | 103 (17) | 292 (14) | 1196 (16) | 372 (31) |
| Health behaviours, n (%) | | | | | |
| Smoking status | | | | | |
| Never smoker | 4453 (38) | 238 (40) | 802 (38) | 2981 (39) | 432 (36) |
| Former smoker | 5937 (51) | 314 (52) | 1121 (53) | 3890 (51) | 612 (51) |
| Current smoker | 789 (7) | 37 (6) | 128 (6) | 530 (7) | 94 (8) |
| Missing | 421 (4) | 18 (3) | 60 (3) | 276 (4) | 67 (6) |
| Alcohol consumption | | | | | |
| Abstainer | 2123 (18) | 78 (13) | 258 (12) | 1479 (19) | 307 (25) |
| Low risk | 3931 (34) | 198 (33) | 742 (35) | 2674 (35) | 317 (26) |
| At risk | 3958 (34) | 238 (40) | 863 (41) | 2566 (33) | 290 (24) |
| Missing | 1588 (14) | 93 (15) | 248 (12) | 958 (12) | 291 (24) |
| Physiological markers*, n (%) | | | | | |
| BMI in kg/m²† | | | | | |
| <23 | 1323 (11) | 107 (18) | 295 (14) | 822 (11) | 99 (8) |

Continued

**Table 1** Continued

| Characteristic | All | 1. Excellent | 2. Good | 3. Moderate | 4. Poor |
|---|---|---|---|---|---|
| ≥23 and < 30$ | 8002 (69) | 436 (72) | 1560 (74) | 5317 (69) | 689 (57) |
| ≥30 | 2264 (20) | 64 (11) | 256 (12) | 1533 (20) | 411 (34) |
| **Blood pressure in mm Hg‡** | | | | | |
| SBP≤140/160 and DBP<90$ | 6888 (59) | 367 (61) | 1271 (60) | 4511 (59) | 739 (61) |
| SBP≤140/160 and DBP≥90 | 92 (<1) | 3 (<1) | 20 (1) | 64 (1) | 5 (<1) |
| SBP>140/160 and DBP<90 | 3822 (33) | 194 (32) | 670 (32) | 2560 (33) | 398 (33) |
| SBP>140/160 and DBP≥90 | 774 (7) | 42 (7) | 145 (7) | 528 (7) | 59 (5) |
| **CHOL/HDL ratio** | | | | | |
| <3.5 | 5561 (48) | 310 (51) | 1040 (49) | 3663 (48) | 548 (45) |
| 3.5–4.9$ | 4540 (39) | 220 (37) | 820 (39) | 3022 (39) | 478 (40) |
| >5 | 1345 (12) | 68 (11) | 227 (11) | 895 (12) | 155 (13) |
| **FEV1/FVC ratio** | | | | | |
| ≥70$ | 8860 (76) | 473 (79) | 1625 (77) | 5862 (76) | 900 (75) |
| <70 | 2740 (24) | 134 (22) | 486 (23) | 1815 (24) | 305 (25) |
| **Physiological markers, n (%)** | | | | | |
| **HbA1c in mmol/mol (% of total Hb)** | | | | | |
| <48 (<6.5%)$ | 9208 (79) | 523 (87) | 1767 (84) | 6072 (79) | 846 (70) |
| 48–52 (6.5%–7%) | 424 (4) | 7 (1) | 43 (2) | 288 (4) | 86 (7) |
| 53–64 (7%–8%) | 324 (3) | 0 (0) | 39 (2) | 217 (3) | 68 (6) |
| >64 (>8%) | 88 (1) | 2 (<1) | 7 (<1) | 57 (1) | 22 (2) |
| **Hb in g/L (mmol/L)§** | | | | | |
| <121/137 (<7.5/8.5)$ | 886 (8) | 46 (8) | 166 (8) | 549 (7) | 125 (10) |
| ≥121/137 (≥7.5/8.5) | 10 545 (91) | 552 (92) | 1921 (91) | 7018 (91) | 1054 (87) |
| **TSH in mIU/L and fT4 in pmol/L** | | | | | |
| TSH: 0.5–4.0 and fT4: 11–19.5$ | 2204 (19) | 99 (16) | 413 (20) | 1466 (19) | 226 (19) |
| TSH>4.0 and fT4≥11 or<11 | 427 (4) | 24 (4) | 61 (3) | 292 (4) | 50 (4) |
| TSH<0.5 and fT4≥11 | 81 (1) | 6 (1) | 8 (<1) | 59 (1) | 8 (1) |
| **eGFR in ml/min/1.73m²¶** | | | | | |
| ≥90$ | 3809 (33) | 179 (30) | 622 (29) | 2568 (33) | 440 (37) |
| 60–89 | 6577 (57) | 375 (62) | 1285 (61) | 4315 (56) | 602 (50) |
| 45–59 | 898 (8) | 40 (7) | 166 (8) | 594 (8) | 98 (8) |
| <45 | 151 (1) | 4 (1) | 14 (1) | 98 (1) | 35 (3) |
| **HSI** | | | | | |
| ≤36$ | 2255 (19) | 128 (21) | 471 (22) | 1486 (19) | 170 (14) |
| >36 | 1502 (4) | 46 (8) | 188 (9) | 1031 (13) | 237 (20) |
| **MMSE score**** | | | | | |
| 25–30$ | 10738 (93) | 552 (92) | 1980 (94) | 7178 (94) | 1028 (85) |
| <25 | 786 (7) | 53 (9) | 122 (6) | 449 (6) | 162 (14) |
| **Sum score physiological markers** | | | | | |
| None affected | 600 (5) | 33 (5) | 132 (6) | 386 (5) | 49 (4) |
| ≤2 | 6606 (57) | 369 (61) | 1298 (61) | 4385 (57) | 554 (46) |
| ≥3 | 4394 (38) | 202 (33) | 670 (32) | 2874 (37) | 589 (49) |

Continued

| Table 1 | Continued | | | | |
|---|---|---|---|---|---|
| Characteristic | All | 1. Excellent | 2. Good | 3. Moderate | 4. Poor |

*Missing percentages for all physiological markers were <1% except for FEV1/FVC ratio (31%); TSH and fT4 (75%); and HSI (68%). Blood-based markers are reported in the International System of Units followed by conventional units if used in database. Values marked with $ are cut-offs used to define normal values.

†Cut-off was adjusted for age.

‡Higher cut-off for SBP was used if participants were aged ≥80.

§Cut-offs are adjusted for sex; men had higher cut-off.

¶Calculated by the Cockcroft Gault formula using serum creatinin in umol/l, age, weight and adjusted for sex.

**Cut-offs are adjusted for level of education.

BMI, body mass index; CHOL, cholesterol; DBP, diastolic blood pressure; eGFR, estimated glomerular filtration rate; FeV1, forced expired volume in one second; fT4, free thyroxine; FVC, forced vital capacity; Hb, haemoglobin; HbA1c, glycated haemoglobin; HDL, high-density lipoprotein; HSI, hepatic steatosis index; MMSE, mini-mental state examination; n, number of participants; SBP, systolic blood pressure; SRH, self-rated health; TSH, thyroid stimulating hormone.

probability of poor SRH trajectory membership relative to the probability of excellent SRH trajectory membership (OR: 10.38; 95% CI: 7.38 to 14.72 for one chronic disease, OR: 37.79; 95% CI 22.35 to 71.75 for two or more chronic diseases). Female gender, low education level, physical inactivity, (former) smoking, alcohol abstinence and presence of three or more abnormal values of physiological markers increased the odds of the probability of poor SRH trajectory membership relative to the probability of excellent SRH trajectory membership (table 4).

Wald tests implied that all trajectory groups were distinguished by the number of self-reported chronic diseases, alcohol consumption and the sum score of affected physiological markers (p values<0.001). However, the results presented in table 4 should be interpreted with caution as all OR calculations are affected by the covariates that were included in the multinomial model to determine the probability of SRH trajectory membership.

Sensitivity analysis including alteration of the composite measure for multimorbidity without anxiety and depressive disorders did not alter trajectory group sizes, shapes and OR (results not shown). Dual trajectory modelling accounting for non-random attrition showed constant annual attrition probabilities between 10% (good SRH) and 17% (poor SRH) for all trajectory groups (online supplementary appendix D and figure D1). Posterior probability of group assignment did not improve when modelling the trajectories accounting for attrition bias (online supplementary appendix D and figure D1).

## DISCUSSION

In this sample of an ongoing large cohort study of Dutch community-dwelling older adults, four stable trajectories of SRH over 5 years were identified. The majority (65.3%) of the participants were classified into the moderate SRH category, followed by good (18.8%), poor (10.2%) and excellent (5.6%) SRH. The results of this study confirmed our a priori hypothesis that the probability of poor SRH trajectory membership was associated with multimorbidity, health risk behaviours and abnormalities in physiological markers. The number of chronic diseases seems to be one of the key factors that determine someone's

probability of SRH trajectory membership, as this was the only covariate under consideration that was significantly associated in all SRH trajectories. In addition, the probability of poor SRH trajectory membership was associated with being women, a low education level, health risk behaviours and presence of three or more affected physiological markers.

Contrary to previous studies investigating trajectories of SRH, this study identified only stable trajectories of SRH of older community-dwelling adults during 5 years.[6–8 45] Other studies with comparable measurement intervals, and study duration identified the majority of their participants in the stable trajectories as well; however they also identified small groups with declining and improving trajectories.[6 8] Sample size was not the limiting factor to identify more groups; however, the posterior diagnostic criteria became worse when adding more than four trajectory groups, indicating four groups was the optimum for our sample. Participants of this study were older than the populations used in other studies investigating trajectories of SRH. Response shift in SRH is known to occur among older adults.[46] Compared with their younger counterparts, older adults are suggested to base their SRH more on psychological and life-style behaviours, and less on functional status and physical health, which might indicate reprioritisation response shift.[47 48] Furthermore, older adults adapt their standards of good health over time, also known as recalibration response shift.[46] In addition, cognitive strategies to accept negative outcomes, as well as someone's beliefs contribute to enhanced levels of well-being, despite negative health outcomes,[49] which can explain the stable trajectories of SRH over time in this study sample.

Consistent with other studies investigating trajectories of SRH, we found strong associations between increasing numbers of baseline self-reported chronic diseases and the probability of poor SRH trajectory membership.[6–8] When participants reported only one chronic disease, they had a two, three-and-half and ten times higher odds of being a member of the good, moderate and poor SRH trajectory compared with the probability of excellent SRH trajectory membership, respectively. People suffering two

**Table 2** Differences between completers and non-completers for baseline variables used in final model

| Characteristic | Completers n=8590 | Non-completers n=3010 | P value |
|---|---|---|---|
| **Demographics** | | | |
| Age in years, median (IQR 25–75)* | 68 (66–72) | 69 (67–73) | <0.001 |
| Male sex, n (%)† | 4132 (48.1) | 1352 (44.9) | 0.001 |
| **Education, n (%)†** | | | |
| Low | 4955 (57.7) | 1608 (53.4) | <0.001 |
| Intermediate | 1678 (19.5) | 359 (11.9) | <0.001 |
| High | 1957 (22.8) | 282 (9.4) | <0.001 |
| Missing percentage | 0.50% | 26% | |
| **Health status** | | | |
| **Self-rated health, n (%)†** | | | |
| Excellent | 551 (6.4) | 94 (3.1) | <0.001 |
| Very good | 1982 (23.1) | 308 (10.2) | <0.001 |
| Good | 5274 (61.4) | 1084 (36.0) | <0.001 |
| Fair | 765 (8.9) | 214 (7.1) | 0.001 |
| Poor | 16 (0.2) | 4 (0.1) | 0.129 |
| Missing percentage | 0% | 43% | |
| **Self-reported chronic diseases, n (%)†** | | | |
| None | 4435 (51.6) | 1641 (54.5) | 0.003 |
| 1 | 3023 (35.2) | 956 (31.8) | 0.004 |
| ≥2 | 1132 (13.2) | 413 (13.7) | 0.399 |
| Missing percentage | 0% | 0% | |
| **Health behaviours** | | | |
| **Physical activity for at least 30 min, n (%)†** | | | |
| ≥5 days/week | 5732 (66.7) | 663 (22.0) | <0.001 |
| 2–4 days/week | 2191 (25.5) | 290 (9.6) | <0.001 |
| ≤1 day/week | 667 (7.8) | 94 (3.1) | <0.001 |
| Missing percentage | 0% | 65% | |
| **Smoking status, n (%)†** | | | |
| Never smoker | 3349 (39.0) | 1104 (36.7) | 0.007 |
| Former smoker | 4628 (53.9) | 1309 (43.5) | <0.001 |
| Current smoker | 613 (7.1) | 176 (5.8) | 0.007 |
| Missing percentage | 0% | 13% | |
| **Alcohol consumption, n (%)†** | | | |
| Abstainer | 1760 (20.5) | 362 (12.0) | <0.001 |
| Low-risk alcohol consumption | 4224 (49.2) | 561 (18.6) | <0.001 |
| At risk alcohol consumption | 2606 (30.3) | 497 (16.5) | <0.001 |
| Missing percentage | 0% | 43% | |
| **Physiological markers†** | | | |
| ≤2 affected | 5859 (68.2) | 1185 (39.4) | <0.001 |
| ≥3 affected | 2731 (31.8) | 1604 (53.3) | <0.001 |

Continued

**Table 2** Continued

| Characteristic | Completers n=8590 | Non-completers n=3010 | P value |
|---|---|---|---|
| Missing percentage | 0% | 7% | |

*Equality of distributions was tested using the Wilcoxon rank-sum test.
†Equality of proportions was tested using the two sample test of proportions.
n, number of participants.

or more self-reported chronic diseases were 38 times more likely for having a higher probability for poor SRH trajectory membership rather than a high probability for excellent SRH trajectory membership. Earlier studies found weaker associations between the probability of poor SRH trajectory membership and the number of chronic diseases.[7 8] The difference in results might be explained by the different number and combinations of covariates used as predictors in different studies. For instance, previous studies focused on chronic physical health disorders to calculate a composite measure of multimorbidity.[6 7] For this study, the 11 most burdensome chronic diseases forecasted for the next decades by the Dutch National Institute for Public Health and the Environment were used to measure chronic diseases, which included depression and anxiety disorders. The inclusion of depression and anxiety disorders in our composite measure of chronic diseases may have led to the strong associations between self-rated chronic diseases and the probability of poor SRH trajectory membership in this study, because depressive symptoms are considered a risk factor for poor SRH.[50] However, sensitivity analyses excluding depression and anxiety disorders in the composite score for chronic diseases led to similar results. Therefore, it is not expected that the differences in composite measures for chronic diseases explain the differences in magnitude of odds for the probability of poor SRH trajectory membership with increasing number of chronic diseases found in this study compared with earlier studies.

Strengths of this study are the large sample size and short measurement intervals for SRH that contribute to the robustness of the findings. In addition, the use of physiological markers next to self-reported data was, to the best of our knowledge, not previously investigated in

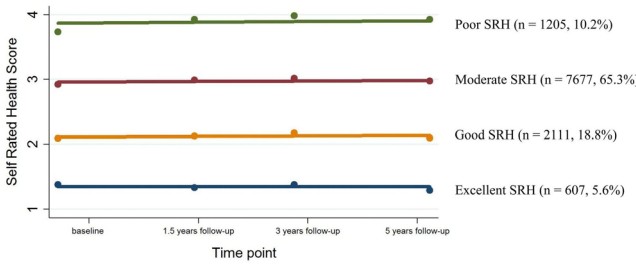

**Figure 1** Non-adjusted trajectories of self-rated health (SRH) over 5 years using 11 600 adults aged 65 years and older of the Lifelines Cohort Study.

**Table 3** Regression estimates (relative risk ratios and 95% CI) of poor SRH relative to excellent SRH from multivariate logistic regression models on SRH trajectory group membership

| Excellent | | Poor SRH trajectory | | |
| --- | --- | --- | --- | --- |
| | | Model 1* | Model 2* | Model 3* |
| Covariate | SRH | n=8679 | n=8679 | n=8590 |
| Age | Ref. | 1.01 (0.99 to 1.04) | 1.02 (0.99 to 1.05) | 1.01 (0.98 to 1.04) |
| Sex | | | | |
| Male | Ref. | Ref. | Ref. | Ref. |
| Female | Ref. | 1.44 (1.09 to 1.90) | 1.66 (1.24 to 2.22) | 1.46 (1.10 to 1.94) |
| Education | | | | |
| Low | Ref. | Ref. | Ref. | Ref. |
| Intermediate | Ref. | 0.76 (0.55 to 1.05) | 0.77 (0.56 to 1.07) | 0.79 (0.57 to 1.10) |
| High | Ref. | 0.50 (0.37 to 0.68) | 0.56 (0.42 to 0.77) | 0.54 (0.40 to 0.74) |
| Chronic diseases | | | | |
| None | Ref. | Ref. | Ref. | Ref. |
| 1 | Ref. | 7.80 (5.74 to 10.61) | 7.03 (5.16 to 9.57) | 7.76 (5.70 to 10.58) |
| ≥2 | Ref. | 26.42 (16.12 to 43.30) | 21.11 (12.80 to 34.82) | 25.08 (15.28 to 41.17) |
| Physical activity for at least 30 min | | | | |
| ≥5 days/week | Ref. | Ref. | Ref. | Ref. |
| 2–4 days/week | Ref. | 1.63 (1.22 to 2.18) | 1.55 (1.16 to 2.08) | 1.61 (1.20 to 2.15) |
| ≤1 day/week | Ref. | 2.82 (1.75 to 4.54) | 2.55 (1.58 to 4.13) | 2.85 (1.76 to 4.59) |
| Smoking status | | | | |
| Never | Ref. | Ref. | Ref. | Ref. |
| Former | Ref. | 1.40 (1.07 to 1.83) | 1.38 (1.05 to 1.80) | 1.39 (1.06 to 1.82) |
| Current | Ref. | 1.71 (1.03 to 2.85) | 1.70 (1.01 to 2.84) | 1.65 (0.98 to 2.78) |
| Alcohol consumption | | | | |
| Abstainer | Ref. | Ref. | Ref. | Ref. |
| Low risk | Ref. | 0.51 (0.36 to 0.71) | 0.53 (0.38 to 0.75) | 0.50 (0.35 to 0.71) |
| At risk | Ref. | 0.48 (0.33 to 0.69) | 0.51 (0.35 to 0.74) | 0.47 (0.33 to 0.69) |
| Abnormal values of physiological markers† | | | | |
| Body composition | Ref. | | 1.35 (1.03 to 1.76) | |
| Cardiovascular function | Ref. | | 1.36 (1.06 to 1.73) | |
| Lung function | Ref. | | 1.12 (0.84 to 1.50) | |
| Glucose metabolism | Ref. | | 3.77 (1.71 to 8.31) | |
| Haematological condition | Ref. | | 1.48 (0.95 to 2.31) | |
| Endocrine function | Ref. | | 0.97 (0.53 to 1.79) | |
| Renal function | Ref. | | 0.74 (0.56 to 0.97) | |
| Liver function | Ref. | | 1.78 (1.16 to 2.74) | |
| Cognitive function | Ref. | | 1.53 (1.00 to 2.34) | |
| Sum score of physiological markers | | | | |
| ≤2 affected | Ref. | | | Ref. |
| ≥3 affected | Ref. | | | 1.51 (1.16 to 1.96) |

Participants with missing data for covariates were excluded from the analyses.
*Fit statistics: Model 1: AIC: 1.807, BIC: −62 488; Model 2: AIC: 1.811, BIC: −61 942; Model 3: AIC: 1.804, BIC: −61 718.
†Participants with normal values of the physiological markers were used as the reference category.
AIC, Akaike information criterion; BIC, Bayesian information criterion; n, number of participants; ref, reference category.

**Table 4** ORs and 95% CI per predictor for being member of the good, moderate or poor SRH trajectory group relative to the excellent group (n=8590*)

| Predictor | OR (95% CI) | | | |
| --- | --- | --- | --- | --- |
| | Exc. SRH | Good SRH | Moderate SRH | Poor SRH |
| | n=471 | n=1716 | n=5637 | n=766 |
| Age | | | | |
| 65–69 | Ref. | Ref. | Ref. | Ref. |
| 70–74 | Ref. | 0.99 (0.75 to 1.33) | 0.93 (0.72 to 1.19) | 1.03 (0.77 to 1.41) |
| 75–79 | Ref. | 1.38 (0.89 to 2.39) | 1.33 (0.88 to 2.18) | 1.34 (0.81 to 2.30) |
| ≥80 | Ref. | 1.15 (0.56 to 2.59) | 1.08 (0.60 to 2.31) | 1.12 (0.56 to 2.78) |
| Sex | | | | |
| Male | Ref. | Ref. | Ref. | Ref. |
| Female† | Ref. | 1.03 (0.76 to 1.39) | 1.21 (0.95 to 1.55) | 1.43 (1.03 to 1.94) |
| Education | | | | |
| Low | Ref. | Ref. | Ref. | Ref. |
| Intermediate† | Ref. | 1.10 (0.78 to 1.53) | 0.87 (0.646 to 1.19) | 0.76 (0.51 to 1.12) |
| High† | Ref. | 0.96 (0.73 to 1.28) | 0.54 (0.41 to 0.68) | 0.47 (0.33 to 0.66) |
| Chronic diseases | | | | |
| None | Ref. | Ref. | Ref. | Ref. |
| 1 | Ref. | 2.11 (1.54 to 2.93) | 3.55 (2.80 to 4.94) | 10.38 (7.38 to 14.72) |
| ≥2 | Ref. | 1.60 (0.92 to 3.30) | 5.29 (3.35 to 10.52) | 37.79 (22.35 to 71.75) |
| Physical activity for at least 30 min | | | | |
| ≥5 days/week | Ref. | Ref. | Ref. | Ref. |
| 2–4 days/week† | Ref. | 0.99 (0.76 to 1.39) | 1.35 (1.08 to 1.80) | 1.61 (1.18 to 2.20) |
| ≤1 day/week | Ref. | 0.95 (0.54 to 1.76) | 1.42 (0.90 to 2.40) | 3.12 (1.76 to 5.16) |
| Smoking status | | | | |
| Never | Ref. | Ref. | Ref. | Ref. |
| Former‡ | Ref. | 1.08 (0.82 to 1.42) | 1.15 (0.91 to 1.44) | 1.48 (1.11 to 1.98) |
| Current† | Ref. | 1.09 (0.66 to 1.95) | 1.42 (0.93 to 2.30) | 1.80 (1.02 to 3.16) |
| Alcohol consumption | | | | |
| Abstainer | Ref. | Ref. | Ref. | Ref. |
| Low risk | Ref. | 1.38 (0.93 to 2.16) | 0.86 (0.62 to 1.19) | 0.52 (0.35 to 0.77) |
| At risk | Ref. | 1.40 (0.97 to 2.12) | 0.78 (0.57 to 1.10) | 0.46 (0.31 to 0.70) |
| Sum score of physiological markers | | | | |
| ≤2 affected | Ref. | Ref. | Ref. | Ref. |
| ≥3 affected | Ref. | 0.89 (0.69 to 1.21) | 1.10 (0.88 to 1.45) | 1.50 (1.14 to 2.03) |

Final trajectory model including identified predictors of SRH trajectory membership by multinomial logistic regression analysis (table 2, model 3) adjusted for age (5 year intervals from 65 years old), education and sex.
*3010 of 11 600 participants aged 65 years and older were excluded from the analysis due to missing data on covariates included in the final model.
†Wald tests showed no differences between poor and moderate SRH trajectories (p>0.05).
‡Wald tests showed no differences between moderate and good SRH trajectories (p>0.05).
Exc, excellent; Ref, reference category; SRH, self-rated health.

combination with trajectory analyses. There were limitations as well. First, although we found a strong association between self-reported diseases and higher probability of poor SRH trajectory membership, we cannot rule out reverse causation. The presented ORs only measure relative change on group level and are not suited to generalise to individual probability of group membership. It is therefore hard to translate these results into concrete clinical implications, as there will always be people having multimorbidity combined with excellent SRH. Second, in this older population, the use of self-reported measurements used for measuring the number of chronic diseases may

have led to an over/under-estimation of the prevalence of diseases due to non-differential misclassification bias. Finally, attrition may have threatened the generalisability of our results.[51] However, sensitivity analysis with trajectories jointly modelled with attrition[52] did not improve group allocation probabilities. In addition, constant annual attrition probabilities below 20% for all groups were identified, which led us to conclude that attrition rates were constant among all trajectory groups.

## IMPLICATIONS AND CONCLUSIONS

This study identified four stable trajectories of SRH over 5 years in Dutch community-dwelling, older adults where the majority of the sample had moderate SRH. Being women, lower levels of education, health risk behaviours (smoking, physical inactivity and alcohol abstinence) and presence of three or more abnormal physiological markers were associated with higher probability of poor SRH trajectory membership. The identified modifiable determinants may provide a basis for future preventive strategies.

**Author affiliations**
¹Department of Internal Medicine and Geriatrics, University of Groningen, University Medical Center Groningen, Groningen, The Netherlands
²Department of Geriatrics, Gelre Hospitals, Apeldoorn, Gelderland, Netherlands
³Department of Internal Medicine, Section of Geriatric Medicine, Academic Medical Center, University of Amsterdam, Amsterdam, The Netherlands
⁴Section of Geriatrics, Department of Internal Medicine, School of Medicine, Yale University, New Haven, Connecticut, USA
⁵Department of Epidemiology, University of Groningen, University Medical Center Groningen, Groningen, The Netherlands

**Acknowledgements** The authors wish to acknowledge the service of the Lifelines Cohort Study and all study participants.

**Contributors** NS obtained funding and supervised the project. MF performed statistical analyses and wrote the first draft of the manuscript. NS and JLM-V aided in interpreting the results. MF, BCvM, JLM-V, SDR and NS were involved in the study design, revising manuscript draft for important intellectual content and gave approval for the final manuscript, and thereby taking full responsibility for the work and manuscript content.

**Funding** This work was supported by the University of Groningen, in collaboration with the University Medical Center Groningen, departments of epidemiology and internal medicine and geriatrics. The Lifelines Biobank initiative was funded by Fonds Economische Structuurversterking, Samenwerkingsverband Noord Nederland and Ruimtelijk Economisch Programma. JLM-V is funded through the Netherlands Organisation for Health Research and Development, grant number 91619060.

**Competing interests** None declared.

**Patient and public involvement statement** This research as well as the Lifelines Cohort Study database development was performed without public or patient involvement.

**Patient consent for publication** Not required.

**Ethics approval** The Lifelines Cohort study was approved by the research ethics committee of the University Medical Center Groningen, The Netherlands (registration number: 2007/152). All participants provided written informed consent before study enrolment.

**Provenance and peer review** Not commissioned; externally peer reviewed.

**Data availability statement** Data are available upon reasonable request. The Lifelines facility is open for all researchers upon request. Information on the application procedure for data access is described on www.lifelines.nl. Researchers interested in queries related to data access may contact the Lifelines Research Office via data@lifelines.nl.

**ORCID iD**
Marlies Feenstra http://orcid.org/0000-0002-1762-7489

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
