## [Reviewer comments · BMJ Open]

ARTICLE DETAILS

TITLE (PROVISIONAL)	Trajectories of self-rated health in an older general population and their determinants: The Lifelines Cohort Study
AUTHORS	Feenstra, Marlies; van Munster, Barbara C; Macneil-Vroomen, Janet L; De Rooij, Sophia; Smidt, Nynke

VERSION 1 – REVIEW

REVIEWER	Hanna Falk Institute of neuroscience and physiology, Sahlgrenska Academy at the University of Gothenburg, Sweden. Institute of health and care sciences, Sahlgrenska Academy at the University of Gothenburg, Sweden.
REVIEW RETURNED	29-Nov-2019

GENERAL COMMENTS	Background  • The aim of this study is to 1) identify classes of self-rated health (SRH) over five years in community- dwelling older adults aged 65 years and older, and 2) to investigate whether group membership of SRH trajectories is associated with self-reported chronic diseases, health risk behaviors, and biomarkers. • In order to do that, the authors suggests using trajectory analysis of latent clusters of individuals who follow a similar pattern of SRH over time. According to the authors, this has not been done previously and could provide important insight into the dynamics underlying SRH in old age. • The authors also point out that in existing studies, objective measures and determinants of health status (for example abnormalities in biomarkers, blood pressure, thyroid hormone levels, and glycated hemoglobin) are not evaluated although they might reflect pre-clinical prodromal phases of underlying diseases. • The authors hypothesize that multi-morbidity, health risk behaviors, and deviations in biomarkers are associated with trajectories that lead to poor SRH. Methods and materials  • 11 600 older adults participated in the study with baseline and three follow-up assessments. • SRH was measured using the standard question with response options ranging from excellent o poor. • Covariates – age, sex, educational attainment, chronic illness (dementia, MI, osteoarthritis CVA, diabetes, COPD, cancer, anxiety, mood disorder/depressed mood), risk behaviors (physical activity, smoking, alcohol consumption)  o Why didn't you add a comorbidity index to measure illness burden?
--

	 o Page 5, line 58 – Please add reference defining “low risk dinking”. • Biomarkers – BMI, blood pressure, HDL, FEV1, FVC, HbA1c, TSH, T4, EGFR, HIS, MMSE. Statistics  • SRH class membership was determined using latent class analysis (GBTM). • This initial step was followed by detailed step-wise description of advanced statistical methods in order to 1) define characteristics of SRH trajectory groups, 2) define covariates of trajectory group membership. o I find the statsitcal description too detailed and would suggest shorting it. All details confuses the reader and makes this section unnecessarily heavy. Results  • Four stable trajectories were identified, including excellent (n = 607, 6%), good (n = 2111, 19%), moderate (n = 7677, 65%), and poor SRH (n = 1205, 10%). • Being female, low education, one or multiple chronic diseases, smoking, physical inactivity, alcohol abstinence, and deviating biomarkers increase the odds for poor SRH trajectory membership compared to excellent SRH trajectory membership. • SRH of community-dwelling older adults aged 65 years and over is stable over time with the majority (65%) having moderate SRH. Older adults reporting poor SRH often have unfavorable health status. o Did you control for situational changes (i.e. the occurrence of an illness or disability/adverse health events) possibly affecting the individual during the five year period? Please clarify this. Especially since SRH did not change between assessments. o Did individuals change group membership over the five year period due to worsening illness, increased physical inactivity, stop drinking alcohol, start smoking, showing deviant biomarkers (for various reasons)? o Page 13, line 5 – Please write out the different groups. • The poor SRH trajectory was characterized by older age, female gender, lower educational attainment, physical inactivity, alcohol abstinence, more chronic illness, higher BMI, higher CHOL/HDL ratio, higher HIS index, lower cognitive function. o Page 13, line 28 – Clarify what you mean by “Hb levels”. o Page 15, line – What is the overall odds of being in the poor group compared to the other groups given that an individual meets all criteria (i.e. female, low education, chronic diseases, smoking, physical inactivity, alcohol abstinence, and deviating biomarkers)? How does this OR change when individuals meet only some of the criteria? o What is the combined effect of all predictors? Interaction effects? You have gone through much trouble identifying the trajectory groups – Make use of these groups and their joint effect on SRH when being a member in one of them. o What is the difference between the factors characterizing the trajectory group and the factors predicting trajectory group membership? Please clarify. Discussion  • The results confirmed the authors’ hypothesis that poor SRH is associated with multi-morbidity, health risk behaviors, and abnormalities in biomarkers, does not change over the five year period, and the number of chronic diseases is a key determinant of SRH.
--	---

	o I am not sure what this adds to the research field. We have known for decades that chronic diseases have a direct effect on SRH and an indirect effect through functional limitations. We also know that the link between symptoms, diagnosed conditions, and functional status on the one hand and SRH on the other hand weakens with age. Could you please emphasize the uniqueness of your study? o Numerous studies have shown that the assessment of SRH is influenced by contextual frameworks of evaluation. It is frequently observed that people have a remarkable ability to adapt to discomfort and illness, and that chronically ill patients generally report levels of HRQoL and SRH higher than one would expect given their condition. In addition to response shift, adaptation (i.e. intrapsychic process in which past, present, and future situations and circumstances are given such cognitive and emotional meaning that an acceptable level of well-being is achieved) could provide important clues to why SRH did not change.
--	---

REVIEWER	Sylvie Bastuji-Garin CEpiA unit (Clinical and Epidemiology and Ageing) (EA 7376) Public Health Department Henri-Mondor Hospital Créteil France
REVIEW RETURNED	29-Nov-2019

GENERAL COMMENTS	This study aimed at assessing whether 5-year trajectories of self-rated health (SRH) are associated with chronic diseases, health risk behaviours, and biomarkers in community-dwelling individuals aged 65 years and older. The question is relevant and the results could be very interesting, but I have several major comments. Major comments  1. The primary outcome may be questionable, indeed can the answer to the following single self-reported question: “how would you rate your health in general?” really reflect the state of health? This is all the more important as the authors show the absence of variations in responses over time, if I have correctly interpreted the figures. So finally it comes down to analysing the factors associated with the baseline SRH. 2. The methods section should be clarified, the authors should better specify the method for identifying trajectories. In particular, the authors should specify whether covariates such as age, sex were used to construct the model. Moreover, the mode of population selection is not clear with discrepancies between different paragraphs of the methods section as well as with the flow diagram, e.g. page 5 “Study population. A subsample of the adult Lifelines Cohort Study was used, including participants aged 65 years or older at baseline (n = 12 685) of which data at baseline and three follow-up measurements over five years period were available.” and page 7 “Data of participants with missing data of were not imputed (n=1085 (9%)) and were therefore excluded from data analyses. Participants with missing data of the main outcome at three or less time points were imputed using maximum likelihood estimation. Thus imputed or excluded? The data of the 3010 (26%) participants who had missing data for baseline covariates were not imputed. Thus the population selection should be clarified as well as the data imputed, only follow-up data? Baseline data? Only covariates or baseline SRH? Furthermore, if numerous (the number should be specified) follow-up SRH data
--

	have been imputed via baseline SRH this could explain the lack of modification of the SRH over time... 3. Discussion. According to the results, this manuscript show factors associated with poor self-rated health rather than factors associated with trajectories. Finally, the most important result of this study is the stability of the SRH over a 5-year period. 4. Tables. Baseline SRH should be added in Table 1 for all SRH trajectories, the number of missing data should be mentioned for each variable in all tables. 5. Figures in appendix. The authors should mention the position index used (mean or median ?) and ad a measure of dispersion 6. Minor points. The term “biomarkers” is not common for blood pressure or MMSE
--	---

VERSION 1 – AUTHOR RESPONSE

Reviewer’s Comments	Our Response (Include here all edits made to the text. For text additions, include verbatim text here in quotes. If no text changes made, provide full justification here)	Location of edits
Reviewer#1		
Methods  • Covariates – age, sex, educational attainment, chronic illness (dementia, MI, osteoarthritis CVA, diabetes, COPD, cancer, anxiety, mood disorder/depressed mood), risk behaviors (physical activity, smoking, alcohol consumption) ➤ Why didn’t you add a comorbidity index to measure illness burden? 	With the available data we were unable to create an existing multimorbidity index. We created our own three level categorical variable (0, 1, ≥2). The variables included in our comorbidity index were: dementia, myocardial infarction, osteoarthritis, cerebrovascular accident (CVA), diabetes, chronic obstructive pulmonary disease (COPD), cancer, anxiety, and mood disorders.	Page 6, lines 50 - 60
 ➤ Page 5, line 58 – Please add reference defining “low risk drinking”. 	We thank the reviewer for identifying this oversight. We added the reference of the NIH guidelines for low-risk drinking and revised the analysis based on the NIH guidelines of max three drinks per day and max seven drinks per week for both men and women. Adjustments were made in Table 1 and Table 2 by adjusting the proportions of participants with low risk and ad risk alcohol consumption; Table 3 by revising all relative risk ratio’s and 95% confidence intervals based on the new limits for ‘low-risk’ and ‘at risk’ drinking behavior; Table 4 and the abstract by revising all OR and 95% CIs; and Table D1 (in Appendix D) by adjusting the posterior probability of group assignments of the revised model. The revised analyses did not change the results nor the conclusions, except for the direction and magnitude of the association between at risk alcohol consumption and the	Page 7, line 10 Page 11, Table 1, lines 14-17 Page 14 Table 2, lines 47-51 Page 16, Table 3, lines 34-37

	probability of moderate SRH trajectory membership relative to the probability of excellent SRH trajectory membership (new OR: 0.78, 95%CI: 0.57; 1.10).	Page 19, Table 4, lines 39-43 Appendix D, page 81, Table D1, lines 14 – 19
Statistics  • SRH class membership was determined using latent class analysis (GBTM). • This initial step was followed by detailed step-wise description of advanced statistical methods in order to 1) define characteristics of SRH trajectory groups, 2) define covariates of trajectory group membership. ➤ I find the statistical description too detailed and would suggest shorting it. All details confuses the reader and makes this section unnecessarily heavy.	We have rewritten and shortened the statistical analysis section in order to make it less detailed but still reproducible by other researchers.	Page 7, lines 55 – 60 Page 8, lines 3 – 60 Page 9, lines 1 – 25
Results ➤ Did you control for situational changes (i.e. the occurrence of an illness or disability/adverse health events) possibly affecting the individual during the five year period? Please clarify this. Especially since SRH did not change between assessments.	The main objective of the current study was first to identify trajectories of repeated measures of SRH over a period of five years. Second, we aimed to identify baseline characteristics that were associated with the probability of group membership of poor, moderate, good and excellent SRH trajectories over time. From a clinical perspective, our interest lied in baseline characteristics associated with probability of SRH trajectory membership, and to a lesser extend with future events. For sensitivity analysis, however, we looked whether the adverse health event mortality during the five year period affected the probability of trajectory membership, which was not the case.	

➤ Did individuals change group membership over the five year period due to worsening illness, increased physical inactivity, stop drinking alcohol, start smoking, showing deviant biomarkers (for various reasons)?	In addition to the previous response, we did not investigate how the probability of group membership changes over time when change in the covariates was observed, as this was beyond the scope of this paper.	
➤ Page 13, line 5 – Please write out the different groups.	We revised the first paragraph of the subheading ‘Trajectories of SRH over 5 years’ in the results section into:	Page 15, Lines 6-15
➤ Page 13, line 28 – Clarify what you mean by “Hb levels”.	‘Of all evaluated models, four trajectories of SRH with different intercepts, and all slopes close to zero showed the best fit (fit statistics are presented in Appendix C Tables C1 and C2). The four trajectories were identified as excellent, good, moderate, and poor SRH including 607 (5.6%), 2111 (18.8%), 7677 (65.3%), and 1205 (9.6%) participants, respectively (Figure 1; Appendix C Figure C1).’ We thank the reviewer for mentioning another oversight. In the method section we added the abbreviation (Hb) to total hemoglobin. We also added ‘higher’ to Hb levels, and HIS index in the result section.	Page 7, Line 24 Page 15, line 33-34

➤ Page 15, line – What is the overall odds of being in the poor group compared to the other groups given that an individual meets all criteria (i.e. female, low education, chronic diseases, smoking, physical inactivity, alcohol abstinence, and deviating biomarkers)? ➤ How does this OR change when individuals meet only some of the criteria? ➤ What is the combined effect of all predictors? Interaction effects? You have gone through much trouble identifying the trajectory groups – Make use of these groups and their joint effect on SRH when being a member in one of them.	We acknowledge that the questions of the reviewer would be of clinical relevance. In addition to the argument on limited space to report all analyses, two substantial arguments were decisive not to expand on individual or combined probabilities of group membership:  1. Because of the great uncertainty involved in these calculations. The odds ratio’s presented in Table 4 in the manuscript were calculated based on the beta’s of the multinomial model to determine the probability of SRH group membership. Consequently, every presented OR is affected by all other variables in the model. We chose not to add the estimated posterior probabilities of group membership per covariate to the paper to prevent incorrect interpretation of the results. 2. Second, the posterior probabilities of SRH group membership will not be the same for individuals with identical scores on the covariates included, as there will always be 	Page 20, lines 8 – 13
---	---	------------------------------

	individuals meeting all criteria (i.e. female, low education, chronic diseases, smoking, physical inactivity, alcohol abstinence, and deviating physiological markers), but who has excellent SRH, and vice versa. Below, we presented a table presenting the probabilities of SRH trajectory membership for different scenarios (e.g. individual and combined effects of all covariates on SRH trajectory membership probability). We would suggest to leave the decision to include this table in the main manuscript to the editor. However, due to the questions of the reviewer, we realized that we should have better emphasized the above motivation in the paper too. Therefore, we added the following phrase about the uncertainty of the OR calculations in the results section: 'However, the results presented in Table 4 should be interpreted with caution as all OR calculations are affected by the covariates that were included in the multinomial model to determine the probability of SRH trajectory membership.' And deleted the following phrase in the sentence 'Table 4 presents... as reference category.': 'independent of the level of other risk factors'	Page 17, lines 43 – 48
➤ What is the difference between the factors characterizing the trajectory group and the factors predicting trajectory group membership? Please clarify.	The current study focused on factors associated with SRH trajectory membership. We revised the results and discussion sections by consistently reporting 'the probability of SRH trajectory membership' throughout the paper when reporting on associated covariates.	Pages 15-23
Discussion  The results confirmed the authors' hypothesis that poor SRH is associated with multimorbidity, health risk behaviors, and abnormalities in biomarkers, does not change over the five year period, and the number of chronic diseases is a key determinant of SRH. I am not sure what this adds to the research field. We have known for decades that chronic diseases have a direct effect on SRH and an indirect effect through functional limitations.	The uniqueness is that we investigated trajectories of SRH over five year and physiological markers associated with the probability of trajectory membership. The results yielded only stable trajectories, which could look like that we 'just' replicated previous studies investigating risk factors of cross-sectionally measured SRH. Nevertheless, the time-aspect should not be overlooked, as the covariates that are previously found to be associated with cross-sectional SRH measurement, were not investigated with trajectories of SRH before (e.g.	Page 23, lines 41 – 43

We also know that the link between symptoms, diagnosed conditions, and functional status on the one hand and SRH on the other hand weakens with age. Could you please emphasize the uniqueness of your study?	physiological markers). We believe that, together with the suggestions the reviewers did, we now better emphasize the uniqueness of the study, that is in the first place having multimorbidity increases the odds for higher probability of poor SRH trajectory membership (described in strengths and limitation section, page 21). Furthermore, the modifiable characteristics low physical activity, smoking, and presence of clinical physiological markers increase the odds for higher probability of poor SRH trajectory membership too. These modifiable risk factors may be target of future preventive strategies..	
➤ Numerous studies have shown that the assessment of SRH is influenced by contextual frameworks of evaluation. It is frequently observed that people have a remarkable ability to adapt to discomfort and illness, and that chronically ill patients generally report levels of HRQoL and SRH higher than one would expect given their condition. In addition to response shift, adaption (i.e. intrapsychic process in which past, present, and future situations and circumstances are given such cognitive and emotional meaning that an acceptable level of well-being is achieved) could provide important clues to why SRH did not change.	We thank the reviewer for the suggestion and added the following phrase to the discussion: 'In addition, cognitive strategies to accept negative outcomes, as well as someone's beliefs contribute to enhanced levels of wellbeing, despite negative health outcomes (48)'	Page 21, lines 33-38
Reviewer#2		
Background The primary outcome may be questionable, indeed can the answer to the following single self-reported question: "how would you rate your health in general?" really reflect the state of health? This is all the more important as the authors show the absence of variations in responses over time, if I have correctly interpreted the figures. So finally it comes down to analyzing the factors associated with the baseline SRH.	The construct of SRH has been valued as a comprehensive, inclusive, and simplistic measure of general health status for a long time (Tissue, 1972; Jylha, 2009). We changed the wording when introducing the concept of SRH in the introduction, and added the reference of Tissue (1972). In the method section, under the paragraph 'primary outcome measure' we added the following phrases and references about psychometric properties of single item SRH question: 'Repeated measures of self-rated health were assessed at...fair, poor) (13,14).' And	Page 5, lines 6 – 8

	“The single item SRH question with five response options is a valid and reliable measure of general health status in older adults (15-17)”	Page 6, lines 29 – 38
Methods ➤ The methods section should be clarified, the authors should better specify the method for identifying trajectories. In particular, the authors should specify whether covariates such as age, sex were used to construct the model.	We have rewritten and shortened the statistical analysis section in order to make it less detailed but still reproducible by other researchers. E.g. we adjusted the first sentence of step 1 as follows to clarify that the basic trajectory model was based on repeated measures of SRH only: ‘Step 1: The basic model was build including the four repeated measures of SRH using a censored normal model.’	Page 8, lines 3 – 6
➤ Moreover, the mode of population selection is not clear with discrepancies between different paragraphs of the methods section as well as with the flow diagram e.g. page 5 “Study population. A subsample of the adult Lifelines Cohort Study was used, including participants aged 65 years or older at baseline (n = 12 685) of which data at baseline and three follow-up measurements over five years period were available.” and page 7 “Data of participants with missing data of were not imputed (n=1085 (9%)) and were therefore excluded from data analyses. Participants with missing data of the main outcome at three or less time points were imputed using maximum likelihood estimation. The data of the 3010 (26%) participants who had missing data for baseline covariates were not imputed. ➤ Thus imputed or excluded? Thus the population selection should be clarified as well as the data imputed, only follow-up data? Baseline data? Only covariates or baseline SRH?	We agree on the reviewer’s comments that the description of the imputed and excluded participants may have led to confusion. We did the following adjustments to the manuscript: We omitted the part “of which data at baseline and three follow-up measurements over five years period were available” We adjusted the following sentences:  - “Data of participants with missing data of the main outcome at all time points were excluded from all analyses (n=1085 (9%)).” - Participants with missing data for the main outcome at three or less time points were handled using maximum likelihood estimation. Maximum likelihood estimation uses all available information from observed data for constructing the likely values for missing data (Nagin, 2005). - From step 2 onwards, participants who had missing data for baseline covariates were excluded from further analyses (n=3010 (26%)). 	Page 6, lines 19 – 20 Page 8, lines 56 – 59 Page 9, line 3 Page 9, lines 8 – 10
➤ Furthermore, if numerous (the number should be specified) follow-up SRH data have been imputed via baseline SRH this could explain the lack of modification of the SRH over time...	Missing data was not imputed but handled using Maximum Likelihood Estimation.	Appendix B, page 77, figure B1, lines 16 – 28

	We adjusted the flowchart in Appendix B by adding:  - The number of missing data for each time point of SRH. - The number of participants excluded for regression analysis. 	
Discussion According to the results, this manuscript show factors associated with poor self-rated health rather than factors associated with trajectories. Finally, the most important result of this study is the stability of the SRH over a 5-year period.	We agree that our main conclusion is that four stable trajectories of SRH were identified over a five-year period, and that given the fact that there were only stable trajectories identified, the results may be extrapolated to risk factors of SRH measures at one time point. However, the multinomial regression analysis were performed to investigate which covariates were associated with the probability of poor, moderate, good, and excellent SRH trajectory membership as dependent outcome. Consequently, the repeated measures design is also incorporated in the multinomial regression analysis. With interpreting the results we tend to focus more on the covariates associated with a higher probability of poor SRH trajectory membership, because poor SRH is known to be associated with negative health outcomes which, in our opinion, makes it clinically the most relevant group.	
Tables  ➤ Baseline SRH should be added in Table 1 for all SRH trajectories 	We thank the reviewer for this remark. Baseline SRH is added to Tables 1 and 2 (first column after 'health status').	Page 10, table 1, lines 19 – 27 Page 14, table 2, lines 18 - 26
 ➤ the number of missing data should be mentioned for each variable in all tables. 	We added the number of missing data for each variable in Table 1. Furthermore, we moved the number of participants used in each regression model from the bottom of Table 3 to the top of the table under the model name. In a similar way, the number of participants allocated to each trajectory group was added to Table 4.	Page 10, table 1 Page 16, Table 3, line 8 Page 19, Table 4, line 8

Figures in appendix ➤ The authors should mention the position index used (mean or median ?) and add a measure of dispersion	We added the following caption to the figure in Appendix D: 'The upper plot represent trajectories of SRH accounted for attrition risk with probability for dropout per trajectory is presented in the lower plot. Dots represent the mean observed value per measurement moment; solid lines represent fit lines; dotted lines in the upper plot represent 95% confidence intervals of the fit lines.'	Appendix D, page 80, capture figure D1, lines 54 – 59
Minor points ➤ The term “biomarkers” is not common for blood pressure or MMSE	We changed the term ‘biomarker’ into physiological marker throughout the manuscript and supplements.	

Table 1: Predicted self-rated health trajectory membership probabilities (95% confidence intervals)

Scenarios	Trajectory membership probability			
	Poor SRH	Moderate SRH	Good SRH	Excellent SRH
1. No risks	.59 (.53; .64)	.27 (.22; .33)	.12 (.11; .13)	.02 (.01; .03)
2. Multimorbidity only	.78 (.69; .94)	.02 (.01; .05)	.00 (.00; .00)	.00 (.00; .00)
3. Physical inactivity only	.79 (.51; .87)	.13 (.07; .23)	.06 (.06; .07)	.01 (.00; .20)
4. Smoking only	.67 (.37; .79)	.18 (.09; .27)	.12 (.10; .13)	.01 (.00; .26)
5. At risk drinking only	.17 (.06; .23)	.20 (.11; .21)	.60 (.28; .66)	.03 (.00; .46)
6. Only deviating physiological markers	.62 (.36; .74)	.20 (.11; .27)	.15 (.12; .16)	.02 (.00; .25)
7. All risk factors	.99 (.94; .99)	.01 (.00; .04)	.00 (.00; .00)	.00 (.00; .02)
Population base rate	.10	.65	.19	.06

All probabilities for each individual risk factor are corrected for age, sex, and educational level. Presented probabilities are affected by all other variables in the model. Confidence intervals are estimated by using the parametric bootstrap technique (1000 replications), explaining their asymmetry.

Abbreviations: SRH, self-rated health

Table 1 reports the probabilities for SRH trajectory membership for seven scenarios. All individual risk factors increase the probability of poor SRH trajectory membership compared to the average population base rate of 10%. Presence of all risk factors shifts the probability of poor SRH trajectory membership to nearly one. However, high uncertainty is involved in these calculations, as there are always individuals having all risk factors that rate their health as excellent and vice versa.